# Effect of Management of Grassland on Prey Availability and Physiological Condition of Nestling of Red-Backed Shrike *Lanius collurio*

**DOI:** 10.3390/ani14071093

**Published:** 2024-04-03

**Authors:** Paweł Knozowski, Jacek J. Nowakowski, Anna Maria Stawicka, Beata Dulisz, Andrzej Górski

**Affiliations:** 1Department of Botany and Evolutionary Ecology, Faculty of Biology and Biotechnology, University of Warmia and Mazury in Olsztyn, Plac Łódzki 3, 10-727 Olsztyn, Poland; anna.stawicka@uwm.edu.pl (A.M.S.); beata.dulisz@uwm.edu.pl (B.D.); 2Department of Zoology, Faculty of Biology and Biotechnology, University of Warmia and Mazury in Olsztyn, Plac Łódzki 3, 10-727 Olsztyn, Poland; andrzej.gorski@uwm.edu.pl

**Keywords:** agriculture, birds ecophysiology, habitat quality, breeding success

## Abstract

**Simple Summary:**

The decline in biodiversity in agricultural areas in much of biological Europe is linked to agricultural intensification and land use changes. Bird species inhabiting natural and semi-natural grassland are among the most endangered ecological groups in Europe. This study aimed to determine the impact of grassland use on the availability of arthropods for red-backed shrike *Lanius collurio* and the condition of their chicks. A total of 133 pastures with different types of use were delineated: 1. extensively used meadows, 2. intensively used meadows fertilised with mineral fertilisers, and 3. intensively used meadows fertilised with manure. The analyses also took into account environmental factors that can affect food availability and condition parameters of chicks (distance to the forest, distance to the river, soil moisture, and proportion of shrubs and tree stands in the meadow border zone). The diversity of arthropods was greatest in extensively used meadows, and the average abundance and proportion of invertebrates over 1 cm in length was highest in intensively used meadows, especially those fertilised with manure. Coefficients characterising body condition parameters (resBCI) and glucose levels of nestlings associated with breeding birds foraging in extensively used meadows indicate good nutrition, but blood parameters suggest severe environmental stress, as is the case in the intensively used meadows fertilised with manure.

**Abstract:**

The study aimed to determine the influence of grassland management on the potential food base of the red-backed shrike *Lanius collurio* and the condition of chicks in the population inhabiting semi-natural grasslands in the Narew floodplain. The grassland area was divided into three groups: extensively used meadows, intensively used meadows fertilised with mineral fertilisers, and intensively used meadows fertilised with liquid manure, and selected environmental factors that may influence food availability were determined. Using Barber traps, 1825 samples containing 53,739 arthropods were collected, and the diversity, abundance, and proportion of large arthropods in the samples were analysed depending on the grassland use type. In the bird population, the condition of the chicks was characterised by the BCI (Body Condition Index) and haematological parameters (glucose level, haemoglobin level, haematocrit, and H:L ratio). The diversity of arthropods was highest in extensively used meadows. Still, the mean abundance and proportion of arthropods over 1 cm in length differed significantly for Orthoptera, Hymenoptera, Arachne, and Carabidae between grassland use types, with the highest proportion of large arthropods and the highest abundance recorded in manure-fertilised meadows. The highest Body Condition Indexes and blood glucose levels of nestlings indicating good nestling nutrition were recorded in nests of birds associated with extensive land use. The H:L ratio as an indicator of the physiological condition of nestlings was high on manure-fertilised and extensively managed meadows, indicating stress factors associated with these environments. This suggests that consideration should be given to the effects of chemicals, such as pesticides or drug residues, that may come from slurry poured onto fields on the fitness of red-backed shrike chicks.

## 1. Introduction

Biodiversity in agricultural areas in much of Europe has been under serious threat for years. Significant declines in the number of species or changes in the community structure of such habitats have been reported for plants, e.g., [1], insects (e.g., [2,3,4]), birds (e.g., [5,6]), and mammals [7]. Declines in biodiversity have been identified as a result of increased agricultural intensification (e.g., [8,9]) and land use changes [10,11,12]. Increased grazing pressure and intensive use of machinery have led to the degradation of soil structure and natural soil renewal processes, resulting in soils of low fertility and biodiversity [13]. Semi-natural grasslands found in temperate climate zones are among the areas with the highest biodiversity in the world [14,15,16]. Meadows of this type are distinguished from other ecosystems by the composition of their flora and fauna and are important habitats for many invertebrates and birds whose occurrence is closely linked to this ecosystem [17,18,19]. At the same time, bird species inhabiting wet meadows are among Europe’s most threatened ecological groups [20,21]. The decline in the number of species and their abundance in agricultural areas is linked to aspects manifested by the intensification of agriculture: the creation of large monocultures, row and field crops, denser and more uniform cropping structures, the loss of native pastures and other semi-natural habitats, an increase in the area of autumn sowings [22,23], or increased use of fertilisers and pesticides [24]. Bird diversity is also negatively affected by the frequency and timing of mowing [25,26]. 

Frequent mowing, homogeneity of vegetation, and excessive use of agrochemicals have contributed to the decline in many pollinators and the loss of habitat for many arthropods [20,27,28], which are the preferred dietary component of many birds [20,29,30,31,32,33]. In the EU, grassland management to halt declines in grassland birds is mainly limited to delaying mowing dates after the chicks have hatched to reduce mortality [34]. The timing of hay harvest is also an important consideration. Moderately late timing of hay harvest conducted simultaneously over large tracts of land is likely to negatively affect the abundance and diversity of the most common arthropods [17,35]. In turn, the cessation of land use leads to changes in species composition and impoverishment of plant communities [36,37], which consequently reduces the availability of invertebrates for birds [20,38]. The diversity, abundance, and availability of invertebrates for birds in restored wetlands are also modified by groundwater levels [39,40,41] and habitat moisture [26,39,42]. In forests and agricultural environments, the negative effects of drought and high temperatures on the abundance of soil fauna, e.g., Collembola, Diptera, and other arthropods, were recorded [43,44,45,46].

Food supplies are usually considered to be a natural and direct factor that can affect bird breeding densities by influencing the survival of full-grown birds or their production of young [47]. Usually, food supply is a critical factor after the young hatch, but food scarcity can also be a serious limiting factor prior to hatching, especially in species that commence breeding in early spring when invertebrate abundance is still low [48] or that have a nestling stage that depends on synchrony with insects’ density [49,50]. A shortage of food during the pre-breeding or egg-laying period can result in a lower proportion of pairs entering breeding [51,52]. In experimental studies with food supplementation, the authors found that the food limitation after the young hatch can also result in a low condition of chicks, high chick mortality, and decreased breeding success [53,54,55]. This may be especially true in anthropogenically modified landscapes [56,57,58]. The availability of food seems to be a key factor for the development of chicks affecting their condition, usually assessed as measurements of morphological traits (e.g., wing length, tarsus, and culmen), body weight, and BCIs (Body Condition Indexes [59,60]), fat score [59,61,62,63], and haematological [59,64,65,66], physiological [67], and biochemical [65,67] parameters. 

The Body Condition Index (body weight divided by one of the body length parameters) is commonly used to assess the condition of birds (see [60,68,69]) and to determine body energy reserves [70,71,72] because the available data suggest that morphometric indices of body condition are often moderately to strongly correlated with fat content in birds [60,73]. Body condition indices have been proven to be a good indicator of the survival prospects of nestling birds [74,75] and to significantly influence their future life [76,77,78]. However, morphometric measurements cannot always effectively detect differences in the condition of birds inhabiting different habitats with consistent access to food [79] or belonging to different populations [80] and often do not reflect the values of other physiological parameters related to bird health or nutritional status [81,82,83]. Measurements based on physiological data, including haemoglobin concentration, haematocrit, haemogram, triglycerides, glucose, and ketone bodies, are commonly used to assess the health of birds [84,85], including in studies of wild birds (e.g., [86,87,88,89,90,91,92,93,94]), and can be used to assess the condition of birds and the effects of environmental stress on their condition [83,95,96,97,98,99]. This is related to the modern view of the condition of the individual [100,101] linked to the influence of many factors shaping weight, metabolic functions, and individual quality. These are influenced by the quality of food (e.g., [102,103,104,105]), infections (e.g., [87,106]), the presence of parasites (e.g., [107,108]) or potentially pathogenic fungi (e.g., [109]), climate (e.g., [110]), genetic and environmental conditions shaping immune function (e.g., [111,112]), and the adaptation of body mass to prey–predator relationships or migration conditions (e.g., [79,113,114,115,116]). All of the above factors contribute to condition variation and influence the costs and benefits of fitness [116,117,118]. In addition to BCIs, measurements of various physiological parameters are therefore used in assessing bird fitness. Minias [98], in a review of more than 120 published studies, concludes that in most of the bird species studied, haemoglobin concentration was positively correlated with commonly used fitness indices such as body weight and fat content, as well as with diet quality. Diet quality is a factor that significantly affects the development of nestlings in the nest, and, as indicated by studies conducted on the Eurasian blue tit *Cyanistes caeruleus* and the great tit *Parus major*, haemoglobin concentration in nestlings is closely related to food quality and availability [64,119]. Haemoglobin concentration in chicks also reliably reflected the intensity of haematophagous ectoparasite invasion [see 98]. Furthermore, haemoglobin concentration correlates with such fitness-related traits as the timing of arrival at breeding grounds, the timing of breeding, egg size, developmental stability, and habitat quality, although these relationships were not always consistent [98].

Another reliable indicator of the impact of environmental stress on vertebrates is the H:L ratio, which changes in response to external stressors, including adverse climatic or weather conditions, environmental pollution, increased reproductive effort, or parasite invasion [99,120]. Baseline levels of the H:L ratio, measured under normal physiological conditions (without acute stress), are highly consistent across individuals over the long term [121], and, hence, the H:L ratio has been most commonly used in physio-ecological studies as a simple but robust measure of physiological stress [96]. Often, changes in the H:L ratio are linked to the response to nutritional conditions [122,123,124,125]. 

Other authors point to glucose and haematocrit as indicators of environmental stress related to both nutrition and pollution. Glucose concentrations may also indicate malnutrition in birds [126]. Haematocrit is one of the more commonly used haematological indices, since, as indicated by a study conducted on the blue tits *Cyanistes caeruleus*, haematocrit is influenced primarily by environmental conditions rather than genetic or maternal factors [127]. On the other hand, Kamiński et al., in studies conducted on white storks, observed the effect of heavy metal element interactions in a polluted environment on Hb (haemoglobin) content and values of red blood [128] and concluded that the use of only Hct (haematocrit) and Hb to assess the health and condition of birds is questionable and gives a positive correlation with various environmental stresses [129]. 

The present work is a continuation of the study of the floodplain of a large lowland central European river (Narew River) included in the Natura 2000 network on the influence of the type and intensity of use of grasslands and their fertilisation regime on the functioning of biodiversity. The Narew and Biebrza river valleys in north-east Poland are among the most valuable areas of biotic diversity in Europe [130]. The flooded river valleys are home to specific meadows that have been extensively used as hay and grazed meadows for many centuries. In Poland, the process of transformation of land use and the intensification of production accelerated significantly in the 1990s. The intensification of agricultural production in north-eastern Poland relates primarily to the increase in cattle numbers and the intensification of grassland management. The transformation of agriculture and the increase in the number of livestock also resulted in an increase in the use of manure and slurry as fertiliser applied to the soil. The intensification of fodder production, resulting in reduced areas of permanent pastures and meadows and more intensive management, may have lowered both the amount of food and its availability to birds. So far, we have shown that the mode of use, especially the frequency of mowing and the type of fertilisation, modified by environmental factors such as distance to the river, soil moisture, and the presence of shrubs affects the biotic diversity of invertebrates, amphibians, and birds [26]. The highest invertebrate diversity was recorded in extensively used meadows introduced in agro-environmental projects, while the strongest reduction in the number of species and their abundance was caused by intensive use of meadows with manure fertilisation [26]. 

The question has arisen as to whether declining invertebrate diversity leads to reduced breeding productivity of insectivorous birds and whether it reduces the condition of nestlings during development in the nest. Stanton et al. [131], in a review paper, report that previous studies have shown negative relationships between agricultural intensification and insect availability, reduced food availability, and lower bird survival and reproduction, but rarely have these two topics been linked. Moreover, the number of papers highlighting the impact of agricultural land use on the physiological aspects of birds is sparse. While the effects of agricultural resource management on bird diversity, species occurrence or abundance, and reproduction have been investigated (e.g., [20,26,29,132,133]), there are few reports on how such factors affect birds’ body or physiological condition during the nesting season (e.g., [134,135,136,137]). 

As a model species, we chose for our study the red-backed shrike *Lanius collurio*, which is a breeding species in agricultural areas in Europe. In western European countries, there has been a significant decline in its population [138,139], which is listed as a species in need of conservation as listed in Annex I of the Birds Directive [Directive 2009/147/EC]. Karlsson, in a study in Finland, shows that a reduction in the heterogeneity and floristic diversity of the herbaceous vegetation (and thus a decrease in the density of invertebrates and their biomass) in intensively cultivated agricultural areas is detrimental to the red-backed shrike due to an increase in the costs necessary for breeding success [140]. Red-backed shrikes feed dependent nestlings mainly with ground-living arthropods (spiders, orthopterans, and beetles) [141,142]. The nestlings are fed food items of mainly medium and large size, and the largest invertebrates are torn apart by adult birds, which feed the chicks with their fragments [142].

This study aimed to analyse how the management of grassland affects the availability of potential food for shrikes and the condition of the chicks of this species. We used the assessment of residuals BCIs and several haematological indices as measures of condition. In addition, we have included in our analyses the influence of selected environmental factors that may shape food availability and chick condition parameters.

## 2. Materials and Methods

### 2.1. Study Area

The study area (Figure 1) is located in north-eastern Poland, in the valley of the Narew River, between the villages of Grądy-Woniecko, Bronowo, Kalinówka-Basie, and Lutostań (endpoints of the study area: EPSG 2180: 1. N: 53°08′23.40″, E: 22°22′54.85″; 2. N: 53°07′15.90″, E: 22°23′41.30″; 3. N: 53°07′42.35″, E: 22°19′02.33″; and 4. N: 53°06′14.14″, E: 22°19′27.93″). The study area had 13.2 km^2^ and was located on humid grasslands in agricultural use. 

The study area was extensively reclaimed in the 1960s [143], and it is currently assigned to the Special Bird Protected Area “Wizna Marsh”. The land drainage investments of the entire Wizna peatland complex, followed by permanent hay-cutting, radically changed the original character of the meadow habitats. In the years 1962–1971, the entire surface of the peat bog was subjected to exhaustive drainage works, and for almost the next 50 years, the dominant areas of swamps, reeds, and shrublands have been replaced by multi-hay meadows with different levels of use. The treatments carried out allowed the clearing of scrub and shrub and a significant increase in the area of grassland used for agriculture [143,144]. The timing of the presence of river flooding during the year and ownership aspects have a key influence on how individual grasslands are managed. The CORINE Land Cover dataset (Copernicus Land Monitoring Service) was used to classify grassland according to one of two soil moisture types: permanently wet areas and temporarily wet areas. There were a total of 133 individual grasslands in the study area, which were of similar size (Figure 1) and for which the type of use (frequency of mowing and type and intensity of fertilisation) was determined. Based on our earlier study [26], three types of meadows were distinguished: extensively used meadows mown once a year (area of approx. 5.3 km^2^), intensively used meadows mown two to four times per year and fertilised with mineral fertiliser (area approx. 5.8 km^2^), and intensively used meadows mown more than twice with cattle slurry (area approx. 2.1 km^2^). In addition, the distance of the centre of each meadow to the riverbed (m) and the edge of the nearest forest (m) and the length of the boundary line of each meadow occupied by shrubs and trees (m) were measured. These variables were measured in “QGIS Desktop 3.4.13” [145] using a loaded orthophoto from “Geoportal 2” (https://mapy.geoportal.gov.pl/ accessed on 1 April 2023), as well as measurements taken in the field using a Leica 1600 rangefinder and a rangefinder integrated with Leica Geovid R 15 × 56 binoculars.

### 2.2. Methods: Arthropods—Field and Laboratory Works

Epigeic arthropods, i.e., the potential food source for the red-backed shrike, were sampled by using Barber pitfall traps. The traps were placed in 27 meadows, along two randomly chosen transects (II and III) across the Narew River valley (Figure 1). Six linearly arranged Barber pitfall traps were placed in each meadow. The traps were filled to about 1/3 of their height with ethylene glycol, using a modified method according to Huruk [146]. Trapping took place in the three growing seasons (May–August) of 2018, 2019, and 2020. Each year, we performed, 4 series of trapping, each one 7 days long, at regular intervals. Therefore, we carried out a total of 12 trapping series. At the end of each trapping series, trap contents were collected from 1944 set-up traps, of which 1825 contained invertebrates and 119 were empty or destroyed by agricultural activities.

In each sample, the identification of invertebrates to orders Arachne, Orthoptera, Hymenoptera, Diptera, Hemiptera, and Coleoptera and, in the case of Coleoptera beetles imago, to families Carabidae, Curculionidae, and Elateridae and their abundances were determined. Large carrion beetles Silphidae were not included in the taxonomic composition of arthropods, as their abundance in the sample was dependent on the number and type of prey present (mainly small vertebrates that accidentally fell into the traps). The Coleoptera larvae also were identified in samples. The taxonomic diversity of arthropods was characterised by the Shannon index.

Arthropods from each sample were spread out on white boards and photographed. Based on the photographs taken, the size of individual specimens was measured to the nearest 1 mm using QGIS Desktop 3.4.13 [145]. A total of 53,739 organisms were measured. The average size of the taxa and the ratio of the number of arthropods over 1 cm in size to the sum of all measured organisms from each taxonomic group were calculated, as in [147,148,149].

### 2.3. Methods: Birds—Field and Laboratory Works

The number of shrike breeding pairs in each breeding season was determined based on detected nests [150]. Surveys were carried out annually between 2017 and 2020 in May–August, during which the territories of pairs of birds were determined using the mapping method [151] and nests were found. During the work, foraging adult birds’ locations were mapped, and data on nest locations and territories were entered into a GPS (Global Positioning System) space using the Garmin GPSmap76S and eTrex Vuista HGx. These data were used to calculate the distances between the closest neighbouring nests using QGis Desktop 3.4.13 software.

Based on the observations of nesting territories and foraging sites of birds, individual nesting territories were assigned to the two most frequently used grasslands. During the study period, we found 145 nests with brood. The average clutch size was 5.0 ± 1.16 eggs/nest and 4.3 ± 0.17 chicks/nest). In extensively used meadows, 32 nests were found (4.9 ± 1.01 eggs/nest; 4.0 ± 1.62 chicks/nest); in the vicinity of meadows fertilised with mineral fertilisers, 70 nests were found (4.9 ± 1.26 eggs/nest, 4.1 ± 1.59 chicks/nest); and in meadows fertilised with manure, 43 nests were found (5.1 ± 1.05 eggs/nest; 4.6 ± 1.14 chicks/nest). The chicks from which samples were taken were approximately 5–10 days old. 

For 97 nests, information related to the body and physiological condition of the 337 chicks was obtained. In chicks, wing length was measured with an accuracy of 1 mm, tarsus length with an accuracy of 0.1 mm, and weight of birds with an accuracy of 0.5 g. Exponential regression models were fitted between body mass and tarsus length (y = −51.905 + exp^(3.911+(0.016*x)^) and between body mass and wing length (y = 6.927^exp(0.074x)^) [152,153]. The residuals from these models (resBC1 and resBCI2, respectively) were used to assess the body condition of chicks. 

Haematological indices (haemoglobin level, haematocrit, and white blood cell indices) and glucose levels were used to assess the physiological condition of the chicks [154,155].

Blood was collected from the cutaneous ulnar vein, in haematocrit microcapillaries, and a blood smear was performed on a basal slide according to, e.g., [156,157]. Haemoglobin [g/dL] was measured using a HemoCue HB 201+ analyser and glucose [mg/dL] using a HemoCue Glucose 201+ analyser. Haematocrit capillaries were centrifuged using a Hettich Haematocrit 200 centrifuge. Haematocrit level (%) was determined from the ratio of the height of the morphotic elements to the height of the whole blood column, which was measured with a Measy electronic calliper to the nearest 0.01 mm. To determine the proportion of leucocyte types in the blood, the “Hemacolor” method was used (Merck, Darmstadt, Germany). The number of heterophils, eosinophils, basophils, monocytes, and lymphocytes was determined in the smears, and the H:L ratio was calculated from these data to assess the level of environmental stress in birds [96].

### 2.4. Statistical Analyses

Relationships between arthropod abundance, arthropod diversity, and the ratio of large arthropods in samples and variation in environmental factors were examined using canonical simultaneous analysis based on three data matrices: the first consisted of the abundance of arthropods in Barber pitfall trap samples collected in individual meadows, the second consisted of the share of arthropods larger than 1 cm in the taxa sample, and the third consisted of environmental variables: distance to the forest boundary (m), distance to the river (m), the length of the meadow boundary line with shrubs and tree stands (m), soil moisture (ranks: 1–3), number of times mown (ranks: 0–4), percentage of mowing area (%), type of meadows use (protected meadows; intensive use, chemical fertilisation; and intensive use, liquid manure fertilisation), and intensity of fertilisation (ranks: 1. lack, 2. moderate fertilisation, 3. excessive fertilisation, and 4. intensive fertilisation). The relationship was analysed using constrained ordination and redundancy analysis—RDA [158]—with Canoco 5 software. The significance of the relation with explanatory variables and the effect of the whole constrained axis were tested using the Monte Carlo permutation test (number of permutations = 99,999). The contribution of individual environmental variables to the explanation of variance in the dependent variables was also tested using the same Monte Carlo permutation test, and probabilities of error of the first type were given for each variable, along with the correction as a result of testing the null hypothesis during multiple comparisons. Corrections were made using the false discovery rate method, a procedure designed to control for the expected proportion of “discoveries” (rejected null hypotheses) that are false. FDR-controlling procedures provide high testing power [159,160].

Both resBCI1 and resBCI2 were highly correlated with each other (0.689, *p* < 0.001; n = 337; Figure A1), so the PCA principal component resulting from both variables (factor loadings = 0.906) was used in further analyses to assess variation in chick body condition depending on environmental factors (hereafter, this new variable is called PCAresBCI1-2). The PCA model and estimation of eigenvalues of the vector were performed using Statistica 13.3 [161].

The relationship of body condition of nestlings and haematological indices (glucose, haemoglobin, haematocrit, and H:L ratio) on environmental variables: distance to the forest boundary (m), distance to the river (m), the length of the meadows boundary line with shrubs and tree stands (m), soil moisture (category: temporary and permanent), type of grassland use (category: 1. lack, protected meadows; 2. intensive, natural fertilisation; and 3. intensive, liquid manure), as well as the number of nestlings per nest and the distance to the nearest neighbouring nest (m), was tested in generalised linear mixed models—GLMMs [162,163]—using IBM SPSS 29 software. Three models were compared in the study. Model A (or initial model) contained only the environmental variables characterising the river valley grassland landscape: distance of the nest to the river, distance of the nest to the forest, soil moisture, length of the border of the meadow covered with shrubs/tree stands, and the number of chicks in the nest. This model was largely unaffected by the factors that characterise grasslands use. Model B (or full model) included all variables from Model A plus those variables characterising the intensity and type of grassland use in the study area and the distance between neighbouring nests. Finally, Model C (or selected model) included only statistically significant explanatory variables. The best-fitting models were selected using the method of introducing the most relevant variables or removing the least relevant in subsequent model-building steps [163,164]. GLMM contained parameters common to all (regression fixed effects and variance components) and cluster-specific parameters estimated from the population distribution. For the variables PCResBCI1-2, glucose level, and haemoglobin level, normal distributions with a link linear function (identity function) were used. For the haematocrit level and H:L ratio, the Gamma distribution and a logarithmic link function were assumed. To exclude some of the variability of the predicted variables (body condition and haematological indexes), the year of bird breeding and the nest ID were included as random variables.

## 3. Results

### 3.1. Diversity of the Potential Food Base of the Red-Backed Shrike

Differences in the composition of potential red-backed shrike prey between the grassland types studied were found for mean arthropod abundance, the proportion of arthropods larger than 1 cm, and arthropod diversity and were modified by several environmental variables linked to the nature of the river valley, which together formed a combination of a system of specific environments.

The environmental factors represented by the first two axes separated by the RDA methods explained 71.38% of the variance in the abundance and diversity of captured invertebrate taxa (Figure 2). The effect of variation in arthropod abundance and diversity variance by the environmental variables studied was statistically significant (pseudo F = 1.7, *p* = 0.045), with two variables playing the strongest role in the variation: distance to the river and number of performed mowing (Table 1). The distance to the river was inversely correlated with soil moisture and determined the more abundant presence of Hemiptera and Curculionidae, and meadows with this location in the space of the Narew valley were the most frequent areas introduced to agri-environmental programmes, extensively used and unfertilised (Figure 2). The diversity of arthropods showed an increase for meadows with extensive type of use (Figure 2). Fertilisation intensity and type of extensive use were the variables influencing the explanation of about 10% of the variance in arthropod diversity and abundance of each (Table 1). The intensively fertilised and mown meadows were characterised by a higher abundance of Elateridae and Orthoptera, partly also Arachne, which was also due to the lower humidity of these habitats located further away from the river (Figure 2).

Large arthropods (greater than 1 cm in length) were most abundant in liquid manure-fertilised meadows, localised at a high distance to the river, with temporary moisture of soil and were mainly large Carabidae, other Coleoptera, and Orthoptera (Figure 3). The proportion of large invertebrates in general was lowest in meadows with an extensive type of use, with large spiders having a higher proportion in this type of grassland (Figure 3). The occurrence of a high proportion of taxa larger than 1 cm was associated with total taxon abundance in the case of Orthoptera, Carabidae, and Curculionidae, whereas for the other studied taxa, especially Arachne, Diptera, Coleoptera larvae, a high proportion of large specimens was rather associated with a specific habitat type and was not associated with total taxa abundance (Figure 3). The factors most strongly influencing the variation in the proportion of large pondweed were the distance to the river, the extensive type of meadow management, the number of mowings per year, the intensity of fertilisation, the degree of soil moisture, and also the percentage of mown meadow area (Table 2).

### 3.2. Variation in Red-Backed Shrike Chick’s Condition between Meadow Types

The body condition of the chicks (PCAresBCI1-2) was higher in nests that were further away from the nests of neighbouring pairs of birds and also depended on the location of the nesting territory to the river and woodland (Table 3). Chick glucose levels were significantly positively related only to brood size (Table 4). Variables entered into the initial model testing the hypothesis of an effect of grassland use intensity on the physiological condition of red-backed shrike chicks were found to be unrelated to variation in their glucose levels (Table 4, Figure 4). Glucose levels were not correlated with indicators of chick body condition (Figure 5). 

The level of haemoglobin in red-backed shrike chicks increased with increasing proximity of nests to the Narew River, as well as in the case of pairs of birds whose nesting territories were located on meadows with a low proportion of shrubs in the birch zone (Table 5). The initial models showed no correlation between haematocrit level and the determined variables (Table 6), while the H:L ratio, like haemoglobin level, increased with decreasing distance from the nest site (Table 7). The addition of variables characterising the intensity of use of the meadows into the A models tested made it possible to conclude that the type of fertilisation and the intensity of use influenced the parameters of the physiological condition of the chicks. The chicks of pairs of birds nutritionally associated with extensively used meadows, like those of birds from manure-fertilised meadows, had low haemoglobin levels compared to birds from territories located on meadows fertilised with mineral fertilisers (Table 5, Figure 4). Haemoglobin level was also strongly related to brood size, increasing significantly in broods with few chicks (Table 5). 

Haematocrit levels depended mainly on the type of grassland use (Table 6, Figure 4); it was significantly higher in chicks associated with mineral-fertilised meadows compared to extensively managed meadows as well as grasslands fertilised with manure (Table 6). Similarly, the H:L ratio was lowest in mineral-fertilised meadows, and the ratio increased in extensively used and manure-fertilised meadows and decreased with increasing distance from the river and woodland (Table 7, Figure 4). Chicks with better body condition had lower H:L ratios (r_s_ = −0.125, n = 282, *p* < 0.001), higher haemoglobin levels (r_s_ = 0.217, n = 306, *p* = 0.0001), and higher haematocrit levels (r_s_ = 0.152, n = 282, *p* = 0.010) (Figure 5).

## 4. Discussion

The transformation of agriculture and the intensification of agricultural production, including the intensification of livestock breeding, is leading to the replacement of extensively used grasslands and pastures with areas of high grass production, resulting in an increase in fertilisation and the use of plant protection products, which are reducing biodiversity. The more intensive management results in a decrease in invertebrate diversity, which is an important food source for many bird species. The red-backed shrike is a predominantly insectivorous species that feeds dependent nestlings mainly with ground-living arthropods (spiders, orthopterans, and beetles) [141,142], preferring large prey [140,141,165] that is not only more easily detectable but has also been shown in other species to have greater nutritional value [166,167].

Karlsson [140] lists Coleoptera (which accounted for 53% of the diet) and Orthoptera (8.3% of the diet) as among the three most common invertebrates found in red-backed shrike pellets. The dominance of beetles and Orthoptera (mainly grasshoppers) in the shrike’s diet is due, among other reasons, to the fact that they are captured by relatively inexpensive energetic methods, known as a ‘sit and wait’ strategy [148,168]. The importance of these two groups of invertebrates in the diet of shrikes is also highlighted by Paczuska et al. [169] who studied the Great Grey Shrike (*Lanius excubitor*) and reported that Coleoptera was found most abundantly in pellets, while Orthoptera dominated in larders. However, it is worth remembering that examining the composition of pellets has its limitations related to the rate of digestion of various groups of insects, which is why the number of identified soft-bodied arthropods is often lower, for example like flies or spider fragments [170]. Kuper et al. [171] indicated that the drastic decline in the abundance of this species in the Netherlands is due to a significant decrease in the number of large invertebrates. Therefore, following the work of Goławski and Goławska [149], we believe that the abundance and proportion of large invertebrates are good indicators of habitat importance for the shrike. 

The highest diversity of arthropods playing a key role in the shrike’s diet throughout its European range [165] was found in extensively managed meadows. Extensively managed meadows had the highest number of Hymenoptera and the highest proportion of large spiders. In a study by Karlsson [140], Hymenoptera accounted for 30.9% of the prey found in pellets and was the second most popular diet source just after beetles. Similar results were reported by Nikolov [141]. On the other hand, soft and non-hard invertebrates such as spiders Araneae, caterpillars Lepidoptera, Opiliones, and Lumbricomorpha appeared two to three times more frequently in the diet of chicks than in that of adult birds [141]. As we have shown in the paper [26], Opiliones in the Narew River valley were strongly correlated in occurrence with soil moisture, which was higher in extensively used meadows close to the river, and their abundance decreased with the intensity of use and fertilisation.

Despite the high diversity of arthropods in extensive meadows, their abundance, which is important for the availability of food for chick-feeding birds, was significantly lower than in more intensively managed meadows. Also, the proportion of large arthropods (greater than 1 cm) was higher in fertilised and intensively managed grasslands than in those extensively managed. Plant biomass, vegetation species structure, and habitat heterogeneity are the main factors affecting the abundance and taxonomic diversity of meadow arthropods [172,173]. Nutrient availability is a major determinant of many ecosystem properties, including primary and secondary production and community structure [174,175]. It can be hypothesised that the higher habitat trophicity that was induced by fertilisation increased the abundance of some invertebrates, especially large insects, but reduced the overall diversity compared to the non-trophic biota. A similar result was obtained by Prather and Kaspari, who indicated in the Pigtail Alley Prairie (Oklahoma, USA) that simply increasing plant biomass through fertilisation increased the abundance, activity, and richness of arthropods [176]. Nutrient loading not only increases plant crop biomass but also plant litter production, which can increase the basal food resource for the detrital community, resulting in increased detritivore and epigeal predator abundances, as predicted by biodiversity and productivity theory, but not affecting species richness [177].

In addition, the use of meadows by mowing them simplifies the habitat structure and most often results in a reduction in arthropod diversity but an increase in arthropod activity, which could move more easily and faster and feed on previously mowed ground [176,177]. Meadows fertilised with mineral fertiliser and slurry were mown several times during the season, resulting in the maintenance of the shortest vegetation. Under short sward conditions, the availability of potential prey for the shrike can increase significantly [178,179]. The role of grasslands with low sward structure is greater than would be suggested by the abundance and biomass of animals in this habitat alone. Moskat [180] showed that longer grass increases the frequency of hunting in flight, which is associated with much higher energy expenditure, so the red-backed shrike prefers to collect prey from the ground when possible. Goławski and Meissner point out that mowing meadows during the breeding season also increases the number of damaged (injured) invertebrates [179]. Therefore, injured and readily available invertebrates are a frequent food of the shrike. The shrike’s use of animals killed during agricultural work in its diet is confirmed by the observations of Van Nieuwenhuyse and Vandekerkhove [181]. The Lesser grey shrike *Lanius minor* also shows a significant increase in foraging intensity in mown meadows compared to unmown meadows [182].

In our study, nestling body condition did not differ significantly between pairs of birds collecting food in extensively managed meadows and intensively managed meadows with different types of fertilisation. Overall, the factor values of the PCAresBCI1-2 component indicated that birds make good use of food for growth during the nesting period in different habitat types regardless of the variation in potential food. The only factor shaping the variation in body condition between nestlings was the aspect of competition, expressed by the influence of the distance of the nearest neighbouring pairs and modified by the distance to the river and forest. The condition of chicks was negatively related to the distance between neighbouring nests. The degree of nourishment of chicks was not associated with the expected negative effect of nesting pair density on chick condition as a result of competition for food resources and reduced food provisioning (e.g., [183,184,185]) and did not decrease with increasing distance between neighbouring nests. Such a result suggests that birds choose certain parts of the environment for the best quality environment for nestling rearing. Despite high body condition indices (BCI), chicks in the extensive meadows had haematological parameters indicative of high levels of environmental stress, as did chicks of pairs that collected food in slurry-fertilised meadows. 

Birds maintain higher plasma glucose concentrations (P(Glu)) than other vertebrates of similar body weight and, in most cases, appear to store relatively little glucose intracellularly as glycogen, with glucose being used as a metabolic substrate [186]. In wild chicks of the Welcome swallow *Hirundo neoxena* and the Spotted dove *Streptopelia chinensis*, Lill et al. reported a developmental increase in BGlu—blood glucose—which probably reflected an increase in metabolic rate as the chicks gained weight and became more active [187]. Baseline glucose levels have also been shown to be an individual characteristic negatively associated with survival (e.g., [42,188]) and to increase as a result of adverse environmental conditions during nestling development [188,189]. Blood glucose and haemoglobin levels may therefore be part of the physiological processes linking environmental conditions to lifespan, a conclusion also drawn from research by Braun and Sweazea [186]. Also, the other two haematological indices studied (haematocrit and H:L ratio) indicate a similar effect of long-term exposure to some stressors on chicks. The effect of stressors that increase the proportion of heterophils and decrease the proportion of lymphocytes present in the blood has been extensively reported in the literature (e.g., [99,120,123,125,127]), and, in addition, birds with a high H:L ratio have been observed to exhibit higher intra-hatch mortality [124] or shortened lifespan and reduced annual survival [190]. Haematocrit, on the other hand, is an indicator of the capacity of the circulatory system to carry oxygen and is related to metabolic levels [191,192].

Many authors have indicated that a reduction in the average haemoglobin level in chicks is associated with limited access to food (e.g., [65,98,193,194]) or poor food quality (e.g., [64,119]). Also, high haematocrit values are correlated with environmental food abundance and the absence of infections and parasites [195,196]. In the case of the Siberian tit *Poecile cinctus*, it was shown that food availability was a stress factor affecting H:L index values [190]. Also, a study by Bańbura et al. [102] involving targeted feeding of Great Tits showed that fed birds have lower H:L index values, while Kaliński et al. [197] suggested that areas with lower food availability for birds may generate chronic environmental stress. 

The results of our study indicate that it is not the availability of food but environmental pollution affecting food quality that may be a factor in the reduced physiological condition of shrike chicks. Environmental pollution with chemical compounds associated with intensive grassland use, particularly chemical compounds associated with liquid manure fertilisation, may trigger physiological responses to adverse factors, particularly under conditions of high metabolism in developing young birds. We believe that an important factor to be analysed in future studies may be the effect of chemicals, which, in the case of our study, may come specifically from the liquid manure poured on the fields. Haematological indices indicate the strongest stress in chicks of birds associated with grasslands of this type of management, but also chicks of birds of extensively managed grasslands included in agri-environmental schemes had similar values of indices indicative of environmental stress, which is an unexpected result. We predict that the surface water run-off in the Narew valley, carrying loads of chemical compounds associated with the use of pesticides and poured manure (e.g., nitrate, antibiotics, and other pharmaceutics), causes their concentration in the extensively managed meadows, as these were located closer to the river in areas that were permanently irrigated, making water-soluble compounds available to plants and subsequently the trophic network of this environment. Antibiotics are widespread and irrationally used in agriculture, medical treatment, and livestock breeding. Most antibiotics ingested by animals are not fully absorbed and metabolised but enter the environment through faeces or the frequent application of faecal or slurry manure, increasing the ecological risk of antibiotics in soil and groundwater. In studies carried out in NE Catalonia in natural springs in an agroecosystem environment with intensive livestock production, high concentrations of nitrate in groundwater, tetracycline residues, and sulphonamides were found, which were characterised by seasonal variations mainly related to hydrological factors and reactive transport processes [198]. Genes conferring resistance to those classes of antibiotics most commonly used in animal production were found in most sources [198].

Pharmaceutics can further be transformed into bioactive compounds, stable and mobile in the environment, with potentially higher toxicity and having a significant impact on living organisms and microbial communities [199]. The negative impact of antibiotic residues on the structure and function of microbial communities is often shown to have the effect of promoting resistance in primary bacteria in the environment, which can be transferred to species associated with infections [200]. It cannot be ruled out that such a risk problem leads to the observed physiological response in chicks. Heterophils are involved in non-specific immune responses; they proliferate into tissues, and their numbers increase in response to prolonged bacterial and fungal infections and abnormalities related to diet and stress [201,202]. The level of stress is not only manifested by an increasing number of heterophils [203,204] but also by a decrease in the number of lymphocytes [205]. It is also manifested by complex responses of the immune system and often compensatory interactions of various components attempting to stabilise the homeostasis of the organism.

## 5. Conclusions

High body condition indices of chicks of pairs of birds feeding on extensively farmed meadows included in agri-environmental programs and a diverse food base indicate favourable conditions for the development of the chicks, but haematological indices describing physiological condition suggest a strong environmental stressor on the chicks. 

Presumably, a stressor could be environmental poisoning by nitrogen compounds and the ingress of pharmaceutical residues through intensive fertilisation of the grassland with slurry, which is reflected in the physiological functioning of the organism and haematological indices indicative of environmental stress.

As the long-term effects of such impacts on the environment cannot be underestimated, our results suggest that further work is required to clarify the condition of the red-backed shrike and other birds associated with this grassland area, particularly as the study area is a Natura 2000 site designated for bird protection and biodiversity. It is important to know if such a condition is widespread in grasslands located on river floodplains.

## Figures and Tables

**Figure 1 animals-14-01093-f001:**
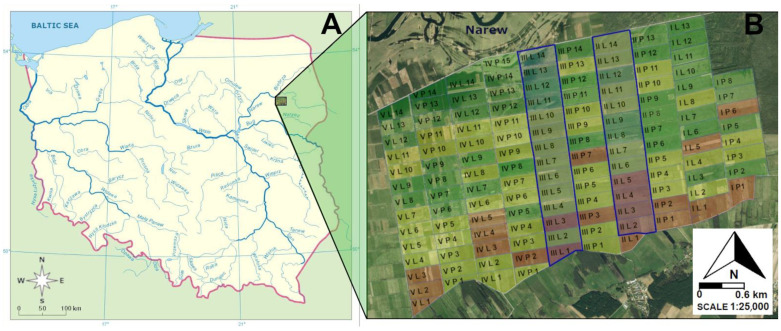
Study area: (**A**)—location of the area in the Narew River valley (NE Poland), (**B**)—different types of meadows use: dark green—extensively used meadows, included in an agri-environmental program; light green—intensively used meadows fertilised with mineral fertiliser; brown—intensively used meadows fertilised with cattle liquid manure; and blue borders—meadows with Barber pitfall traps.

**Figure 2 animals-14-01093-f002:**
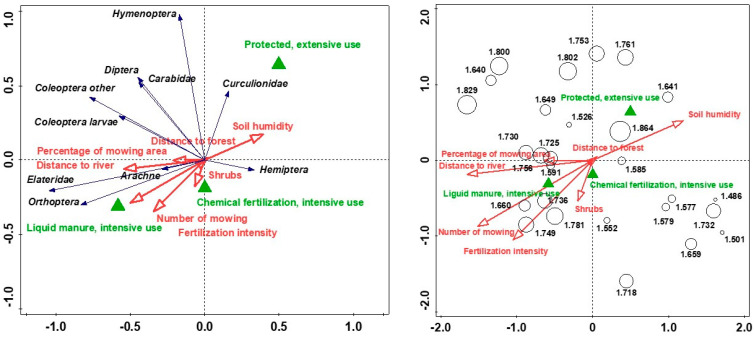
The arthropod abundance—environmental variables (**left**) and arthropod diversity (Shannon index)—environmental variables (**right**) biplots of RDA (the first two axes) in the study area; the two axes explained 71.38% of the total variance.

**Figure 3 animals-14-01093-f003:**
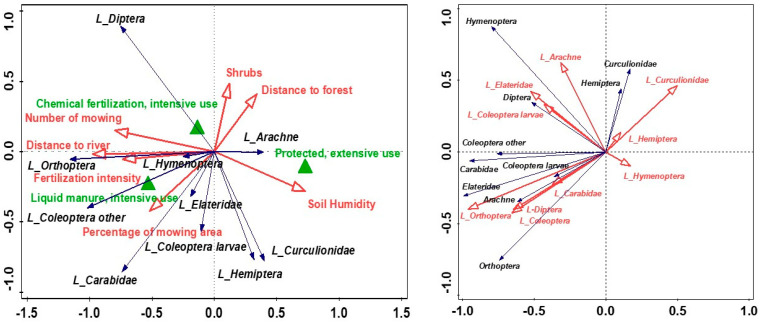
The share of large arthropods—environmental variables (**left**) and taxon’s abundance—share of large arthropods (**right**) biplots of RDA (the first two axes) in the study area; the two axes explained 60.56% and 77.73% of the total variance, respectively.

**Figure 4 animals-14-01093-f004:**
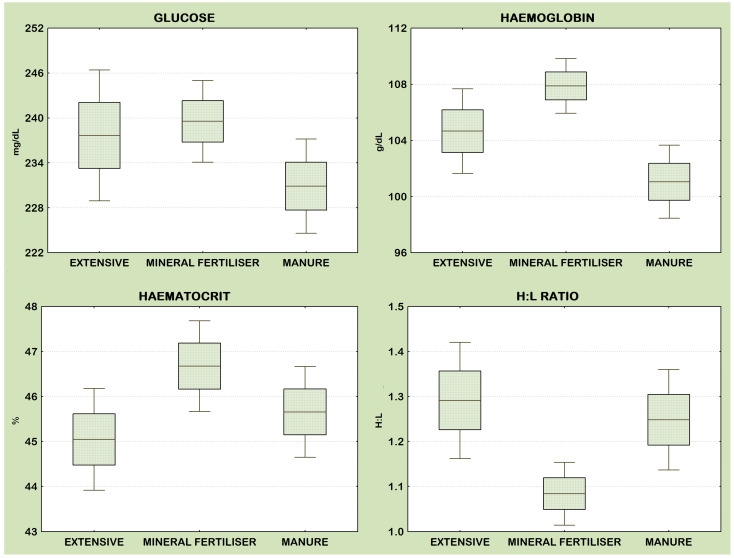
Variation in haematological parameters of red-backed shrike chicks between grassland types. Box plot: line mean, box—95% C.I.; whiskers—min–max.

**Figure 5 animals-14-01093-f005:**
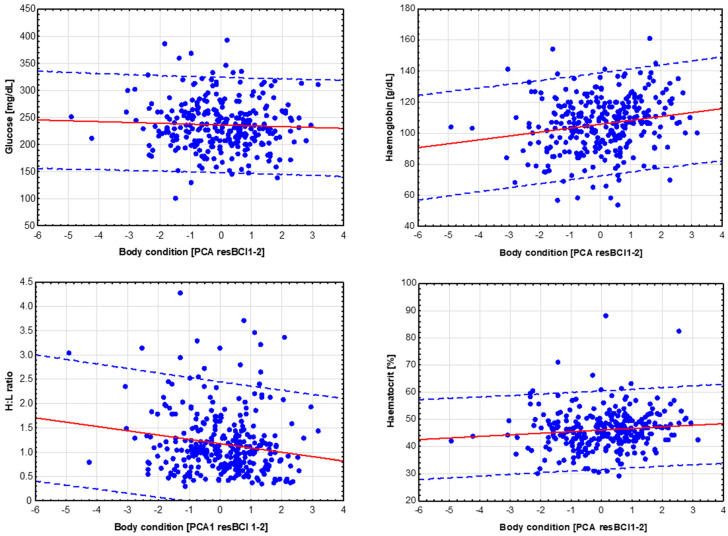
Relationship between body condition and haematological parameters of red-backed shrike chicks; striped lines—95% predicted.

**Table 1 animals-14-01093-t001:** Effects of individual environmental variables on variation in arthropod abundance and diversity. Explains%—the percentage of variance explained; pseudo F—the value of permutation F-test; *p*—probability; and *p*(*adj.*)—probability adjusted for the number of variables compared.

Effects	Explains %	Pseudo-F	*p*	*p*(*adj.*)
**Distance to river**	**15.3**	**4.5**	**0.006**	**0.034**
**Number of mowings**	**15.0**	**4.4**	**0.007**	**0.034**
Intensity of fertilisation	10.3	2.9	0.036	0.115
Type of use: protected, extensive use	9.3	2.6	0.049	0.115
Soil humidity	9.0	2.5	0.057	0.115
Type of use: manure	7.5	2.0	0.098	0.164
Distance to forest	4.8	1.3	0.265	0.379
Percentage of mowing area	3.3	0.9	0.462	0.577
Type of use: chemical fertilisation, intensive use	2.9	0.8	0.529	0.588
Shrubs	2.5	0.6	0.688	0.688

**Table 2 animals-14-01093-t002:** Effect of individual environmental variables on variation in the proportion of large arthropods. Explains%—the percentage of variance explained; pseudo F—the value of permutation F-test; *p*—probability; and *p*(adj.)—probability adjusted for the number of variables compared.

Effects	Explains %	Pseudo-F	*p*	*p*(*adj.*)
**Distance to river**	**13.8**	**4.0**	**0.00007**	**0.0007**
**Type of use: protected, extensive use**	**11.5**	**3.3**	**0.0011**	**0.004**
**Number of mowings**	**11.0**	**3.1**	**0.0013**	**0.004**
**Intensity of fertilisation**	**10.5**	**2.9**	**0.002**	**0.006**
**Soil humidity**	**9.6**	**2.7**	**0.005**	**0.011**
**Percentage of mowing area**	**8.1**	**2.2**	**0.022**	**0.037**
Distance to forest	5.6	1.5	0.151	0.216
Type of use: chemical fertilisation, intensive use	4.7	1.2	0.270	0.318
Type of use: manure	4.6	1.2	0.286	0.318
Shrubs	4.4	1.2	0.341	0.341

**Table 3 animals-14-01093-t003:** Generalised linear mixed models explaining the dependence of the body condition of red-backed shrike chicks (PCAresBCI1-2) on environmental factors in the study area.

A—initial model: F = 0.627, *p* = 0.679; dependent: normal, link-function–log					
**Effect**	**Level of Effect**	**Parameter**	**95% C.I. of Parameter**	**t Statistic**	** *p* **
Intercept		−0.998	−2.378–0.383	1.420	0.156
Number of chicks in nest		−0.013	−0.149–0.123	0.189	0.850
Distance to river		0.0003	−0.0001–0.001	1.450	0.148
Distance to forest		0.0003	−0.0002–0.001	1.294	0.196
Shrubs		−0.0002	−0.001–0.001	0.636	0.525
Soil humidity	Permanently	−0.085	−0.429–0.259	0.483	0.629
Random effect (Year)		0.000			
Random effect (Nest)		0.391	0.297–0.515	Z = 7.102	<0.001
B—full model: F = 5.431, *p* < 0.001; dependent: normal, link-function–log					
**Effect**	**Level of Effect**	**Parameter**	**95% C.I. of Parameter**	**t Statistic**	** *p* **
**Intercept**		**−219.224**	**−361.043–−77.404**	**3.037**	**0.003**
**Distance to nearest nest**		**−0.002**	**−0.003–0.001**	**5.745**	**<0.001**
Distance to river		0.0001	0.0002–0.0005	0.883	0.377
Distance to forest		0.0003	0.0002–0.001	1.297	0.195
Soil humidity	Permanently	−0.175	−0.468–0.118	1.174	0.241
Grassland use type	Manure	−0.141	−0.475–0.193	0.829	0.408
	Mineral fertiliser	0.076	−0.180–0.331	0.580	0.562
Number of chicks in nest		−0.033	−0.114–0.048	0.800	0.424
Shrubs		−0.0002	−0.001–0.001	0.453	0.651
Random effect (Year)		0.018			
Random effect (Nest)		0.057	0.033–0.097	Z = 3.661	<0.001
C—selected model: F = 3.080, *p* = 0.016; dependent: normal, link-function–log					
**Effect**	**Level of Effect**	**Parameter**	**95% C.I. of Parameter**	**t Statistic**	** *p* **
**Intercept**		**593.807**	**172.761–1014.852**	**2.771**	**0.006**
**Distance to nearest nest**		**−0.003**	**−0.005–−0.001**	**2.605**	**0.009**
**Distance to river**		**0.001**	**0.0001–0.002**	**2.249**	**0.025**
**Distance to forest**		**0.002**	**0.0004–0.003**	**2.732**	**0.007**
Soil humidity	Permanently	−0.298	−0.873–0.277	1.017	0.309
Random effect (Year)		0.092	0.014–0.606	Z = 1.037	0.300
Random effect (Nest)		1.032	0.657–1.622	Z = 4.340	<0.001

**Table 4 animals-14-01093-t004:** Generalised linear mixed models explaining the dependence of the glucose level of red-backed shrike chicks on environmental factors in the study area.

A—initial model: F = 1.804, *p* = 0.111; dependent: normal, link-function–log					
**Effect**	**Level of Effect**	**Parameter**	**95% C.I. of Parameter**	**t Statistic**	** *p* **
**Intercept**		**48.396**	**17.842–78.949**	**3.113**	**0.002**
**Number of chicks in nest**		**0.031**	**0.009–0.053**	**2.795**	**0.005**
Distance to river		−0.00001	−0.00008–0.0001	0.352	0.725
Distance to forest		−0.00002	−0.0001–0.001	0.517	0.606
Shrubs		0.00002	−0.0001–0.0001	0.535	0.593
Soil humidity	Temporary	0.013	−0.031–0.057	0.588	0.557
Random effect (Year)		0.001	0.00002–0.012	Z = 0.625	0.532
Random effect (Nest)		0.017	0.011–0.027	Z = 4.227	<0.001
B—full model: F = 1.030, *p* = 0.412; dependent: normal, link-function–log					
**Effect**	**Level of Effect**	**Parameter**	**95% C.I. of Parameter**	**t Statistic**	** *p* **
**Intercept**		**9581.887**	**2215.528–16948.245**	**2.556**	**0.011**
Distance to nearest nest		0.003	−0.015–0.022	0.345	0.730
Distance to river		−0.002	−0.019–0.015	0.785	0.377
Distance to forest		−0.005	−0.027–0.017	0.431	0.667
Soil humidity	Permanently	−3.103	−14.054–7.847	0.557	0.578
Grassland use type	Manure	−1.128	−15.168–12.911	0.158	0.875
	Mineral fertiliser	2.517	−9.327–14.362	0.418	0.676
**Number of chicks in nest**		**6.809**	**1.471–12.148**	**2.507**	**0.013**
Shrubs		0.005	−0.013–0.024	0.589	0.556
Random effect (Year)		25.002	1.000–624.801	Z = 0.609	0.543
Random effect (Nest)		969.863	606.631–1550.587	Z = 4.177	<0.001
C—selected model: F = 4.239, *p* = 0.015; dependent: normal, link-function–log.					
**Effect**	**Level of Effect**	**Parameter**	**95% C.I. of Parameter**	**t Statistic**	** *p* **
**Intercept**		**47.939**	**17.848–78.031**	**3.131**	**0.002**
**Number of chicks in nest**		**0.031**	**0.009–0.053**	**2.811**	**0.005**
Shrubs		0.00002	−0.00005–0.0001	0.463	0.643
Random effect (Year)		0.001	0.00002–0.012	Z = 0.626	0.531
Random effect (Nest)		0.017	0.011–0.026	Z = 4.240	<0.001

**Table 5 animals-14-01093-t005:** Generalised linear mixed models explaining the dependence of the haemoglobin level of red-backed shrike chicks on environmental factors in the study area.

A—initial model: F = 3.300, *p* = 0.006; dependent: normal, link-function–linear					
**Effect**	**Level of Effect**	**Parameter**	**95% C.I. of Parameter**	**t Statistic**	** *p* **
Intercept		581.668	−916.524–2079.860	0.763	0.446
Number of chicks in nest		−1.446	−3.437–0.546	1.426	0.154
Distance to river		−0.005	−0.011–0.0002	1.891	0.059
Distance to forest		−0.004	−0.012–0.003	1.150	0.251
**Shrubs**		**−0.009**	**−0.015–0.003**	**2.856**	**0.004**
Soil humidity	Permanently	1.506	−2.291–5.303	0.779	0.436
Random effect (Year)		0.193	0.000004–9279.117	Z = 0.182	0.856
Random effect (Nest)		172.522	111.682–266.503	Z = 4.507	<0.001
B—full model: F = 2.354, *p* = 0.017; dependent: normal, link-function–linear					
**Effect**	**Level of Effect**	**Parameter**	**95% C.I. of Parameter**	**t Statistic**	** *p* **
Intercept		834.189	−982.030–2650.407	0.903	0.367
Distance to nearest nest		−0.002	−0.009–0.005	0.528	0.597
**Distance to river**		**−0.006**	**−0.013–−0.0002**	**2.032**	**0.043**
Distance to forest		−0.006	−0.014–0.002	1.446	0.149
Soil humidity	Permanently	1.335	−2.497–5.168	0.685	0.494
Grassland use type	Manure	1.971	−2.911–6.853	0.793	0.428
	Mineral fertiliser	−0.442	−4.352–3.467	0.222	0.824
Number of chicks in nest		−1.619	−3.676–0.437	1.547	0.122
**Shrubs**		**−0.008**	**−0.014–−0.002**	**2.563**	**0.011**
Random effect (Year)		0.330	0.0002–690.006	Z = 0.256	0.798
Random effect (Nest)		182.941	117.514–284.797	Z = 4.428	<0.001
C—selected model: F = 7.240, *p* = <0.001; dependent: normal, link-function–linear					
**Effect**	**Level of Effect**	**Parameter**	**95% C.I. of Parameter**	**t Statistic**	** *p* **
**Intercept**		**2187.001**	**619.360–3754.643**	**2.741**	**0.006**
**Number of chicks in nest**		**−1.328**	**−2.521–−0.135**	**2.186**	**0.029**
**Grassland use type**	Manure	−1.309	−5.541–2.922	0.608	0.544
	**Mineral fertiliser**	**4.966**	**1.215–8.718**	**2.601**	**0.010**
Random effect (Year)		1.217	0.051–29.280	Z = 0.616	0.538

**Table 6 animals-14-01093-t006:** Generalised linear mixed model explaining the dependence of the haematocrit level of red-backed shrike chicks on environmental factors in the study area.

A—initial model: F = 0.685, *p* = 0.635; dependent: Gamma, link-function–log					
**Effect**	**Level of Effect**	**Parameter**	**95% C.I. of Parameter**	**t Statistic**	** *p* **
Intercept		−22.558	−47.679–2.563	1.765	0.078
Number of chicks in nest		−0.010	−0.029–0.009	1.051	0.294
Distance to river		−0.00001	−0.0001–0.00005	0.298	0.766
Distance to forest		−0.00002	−0.0001–0.0001	0.547	0.585
Shrubs		−0.0000	−0.0001–0.00002	1.237	0.217
Soil humidity	Permanently	0.014	−0.028–0.055	0.656	0.512
Random effect (Year)		0.0002	0.00001–0.006	Z = 0.574	0.566
Random effect (Nest)		0.011	0.007–0.017	Z = 4.142	<0.001
B—full model: F = 1.432, *p* = 0.181; dependent: Gamma, link-function–log					
**Effect**	**Level of Effect**	**Parameter**	**95% C.I. of Parameter**	**t Statistic**	** *p* **
**Intercept**		**−24.661**	**−42.093–−7.228**	**2.780**	**0.006**
Distance to nearest nest		0.00004	−0.00001–0.0001	1.493	0.136
Distance to river		0.00002	−0.00002–0.0001	0.877	0.376
Distance to forest		0.00001	−0.00005–0.0001	0.337	0.736
Soil humidity	Permanently	0.009	−0.033–0.052	0.440	0.660
**Grassland use type**	Manure	0.027	−0.026–0.080	1.015	0.311
	**Mineral fertiliser**	**0.054**	**0.011–0.096**	**2.468**	**0.014**
Number of chicks in nest		0.002	−0.011–0.015	0.262	0.794
Shrubs		−0.0001	−0.0001–0.00001	1.724	0.085
Random effect (Year)		0.0002	0.00001–0.005	Z = 0.519	0.798
C—selected model: F = 2.514, *p* = 0.058; dependent: Gamma, link-function–log					
**Effect**	**Level of Effect**	**Parameter**	**95% C.I. of Parameter**	**t Statistic**	** *p* **
**Intercept**		**−24.382**	**−41.462–−7.302**	**2.806**	**0.005**
Distance to nearest nest		0.00003	−0.00002–0.0001	1.311	0.191
**Grassland use type**	Manure	0.034	−0.013–0.080	1.414	0.158
	**Mineral fertiliser**	**0.051**	**0.010–0.091**	**2.455**	**0.014**
Random effect (Year)		0.0002	0.00001–0.004	Z = 0.646	0.519

**Table 7 animals-14-01093-t007:** Generalised linear mixed models explaining the dependence of the H:L ratio of red-backed shrike chicks on environmental factors in the study area.

A—initial model: F = 1.280, *p* = 0.271; dependent: Gamma, link-function–log					
**Effect**	**Level of Effect**	**Parameter**	**95% C.I. of Parameter**	**t Statistic**	** *p* **
**Intercept**		**135.551**	**54.809–216.293**	**3.299**	**0.001**
Number of chicks in nest		0.034	−0.022–0.089	1.186	0.236
**Distance to river**		**−0.0002**	**−0.0004–−0.00004**	**2.376**	**0.018**
Distance to forest		−0.0002	−0.0004–0.00002	1.722	0.077
Shrubs		0.00001	−0.0002–0.0002	0.051	0.960
Soil humidity	Permanently	0.002	−0.123–0.126	0.024	0.981
Random effect (Year)		0.005	0.0002–0.101	Z = 0.646	0.518
Random effect (Nest)		0.091	0.056–0.148	Z = 4.025	<0.001
B—full model: F = 3.714, *p* < 0.001; dependent: Gamma, link-function–log					
**Effect**	**Level of Effect**	**Parameter**	**95% C.I. of Parameter**	**t Statistic**	** *p* **
**Intercept**		**180.928**	**125.257–−236.600**	**6.386**	**<0.001**
Distance to nearest nest		−0.000001	−0.0002–0.0002	0.017	0.987
**Distance to river**		**−0.0002**	**−0.0004–−0.0001**	**3.132**	**0.002**
**Distance to forest**		**−0.0003**	**−0.0005–−0.0001**	**3.547**	**<0.001**
Soil humidity	Permanently	0.004	−0.123–0.130	0.055	0.956
**Grassland use type**	Manure	−0.033	−0.191–0.126	0.404	0.686
	**Mineral fertiliser**	**−0.198**	**0.328–−0.069**	**3.015**	**0.003**
**Number of chicks in nest**		**0.041**	**0.001–0.080**	**2.028**	**0.043**
**Shrubs**		0.00005	−0.0002–0.0003	0.426	0.671
Random effect (Year)		0.008	0.0005–0.140	Z = 0.690	0.490
C—selected model: F = 3.619, *p* < 0.001; dependent: Gamma, link-function–log					
**Effect**	**Level of Effect**	**Parameter**	**95% C.I. of Parameter**	**t Statistic**	** *p* **
**Intercept**		**228.407**	**160.305–296.509**	**6.594**	**<0.001**
Number of chicks in nest		0.040	−0.008–0.088	1.632	0.103
Distance to nearest nest		0.00004	−0.0001–0.0002	0.456	0.649
**Grassland use type**	Manure	−0.055	−0.248–0.137	0.565	0.573
	**Mineral fertiliser**	**−0.229**	**−0.387–−0.071**	**2.843**	**0.005**
**Distance to river**		**−0.0002**	**−0.0004–−0.00004**	**2.441**	**0.015**
**Distance to forest**		**−0.0004**	**−0.001–−0.0001**	**3.350**	**<0.001**
Soil humidity	Permanently	0.023	−0.132–0.177	0.286	0.775
Random effect (Year)		0.013	0.001–0.220	Z = 0.690	0.490

## Data Availability

The primary data supporting the results of this study are available upon a reasonable request to the corresponding authors.

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
