# Peer review of "Effect of Management of Grassland on Prey Availability and Physiological Condition of Nestling of Red-Backed Shrike Lanius collurio"

_animals, 2024, doi:10.3390/ani14071093_

Round 1
Reviewer 1 Report
Comments and Suggestions for Authors
Author Response
Jacek J. Nowakowski,
Department of Botany and Evolutionary Ecology,
Faculty of Biology and Biotechnology
University of Warmia and Mazury in Olsztyn
10-727 Olsztyn, Plac Lodzki 3, Poland
Phone: +48 89 523 43 43
e-mail: [email protected]
Responses to Reviewer 1
Dear Reviewer,
Thank you very much for your review. We are pleased that you shared our idea of presenting the collected and processed data on the body and physiological condition of red-backed shrike chicks against the background of potential food resources and, above all, methods of managing grasslands in the river flood valley. Thank you very much for pointing out the linguistic inaccuracies. All comments were taken into account and the corrections were applied to the text. We also agreed with the suggestions to remove several threads in the Introduction (lines 141-144, 146-149, 158-167) and Discussion (lines 498-503). These threads have been removed, which will undoubtedly contribute to better communicativeness of the work, and also allow us to slightly shorten the already quite long text. The manuscript currently also contains some corrections to take into account the comments of Reviewer No. 2. Once again, I would like to thank you for your effort in reading this work.
Kind regards
Jacek J. Nowakowski
Reviewer 2 Report
Comments and Suggestions for Authors
This is an important and interesting paper that broadly investigates the impacts of agricultural intensification on biodiversity in central Europe. The study documents the ecology and physiology of breeding red-backed shrikes across three treatments of meadow intensification in the valley of the Narew River in Poland. Importantly, in addition to looking at invertebrate composition of meadows and reproductive success of breeding shrikes, it also looks at measures of body condition and stress in adults and chicks. Rarely have these diverse measures been combined in a coherent study.
Below I provide a small number of comments. I have also annotated a copy of the manuscript - see attached. I have indicated minor changes in spelling that are needed. The standard of the English is very good; however, I think the suggested changes are worth making.
Introduction. The Introduction is currently long being nine paragraphs in length with some paragraphs being 30 lines long. The Introduction is very informative and useful to set up the key topics for the study but I consider that it is too long and reads more like a literature review. The paragraphs on indicators of environmental stress, for example, could be reduced and be more focussed. I have suggested some text that can be deleted without a significant loss of information.
Line 93-97. This sentence needs rewriting. In its current form there is too much detail and it is confused at the end.
Line 127. The common name given here - praying blue tit – must be a mistake? Eurasian blue tit is the common name I’m most familiar with. Can you please clarify this?
Line 231. Can you please describe – in a few words – the meaning of the terms ‘shrub’ and ‘scrub’? I am not clear on what type of vegetation is involved and how the two types differ.
Line 285. What does pitch mean here? Is it wing length?
Line 493. I think that this word should be ‘injured’ rather than ‘uninjured’.
Lines 498-503. This paragraph seems unnecessary. It is a single sentence and is covered in enough detail in the subsequent text.
Lines 515-516. Here competition is invoked but it seems that there is a sufficient supply of food. Over what resources does intraspecific competition occur? If no resource is limited then it is not wise to invoke competition.

Comments on the Quality of English LanguageThe standard of the English is very good; however, it can be improved. I have suggested changes in a number of places - see annotated file attached.
Author Response
Jacek J. Nowakowski,
Department of Botany and Evolutionary Ecology,
Faculty of Biology and Biotechnology
University of Warmia and Mazury in Olsztyn
10-727 Olsztyn, Plac Lodzki 3, Poland
Phone: +48 89 523 43 43
e-mail: [email protected]
Responses to Reviewer 2
Dear Reviewer,
Thank you very much for your review and the positive opinion it contains about our research and the manuscript submitted for evaluation to Animals. We would like to thank you for all your insightful comments on the manuscript, as well as for pointing out some shortcomings of this message and the need for certain explanations, additions, and corrections. Below, point by point, we present answers and explanations, as well as information about what was included in the manuscript
Introduction
L62: “For most authors…”, consider rephrasing. Most authors are likely not experiencing declines in biodiversity, the grassland communities are. Or studies show..
Answer:
Of course, it has been corrected as suggested. In the current version, it is line 66.
L70-71: Based on cited literature, arthropods are the preferred dietary component of many birds in grasslands. This is also true for most birds in other habitats, which you could broaden the context here by citing additional research for support.
Answer:
Dear Reviewer, this is of course true, this thread can be expanded with information about other types of habitats. But the work is very long these days. The second Reviewer even pointed out some threads that needed to be removed. In this case, we decided to stick only to this perspective on grassland. We refer to over 200 sources and we did not want to expand this thread yet. I hope this will be acceptable to You. If it is necessary, we are of course also ready to expand this thread and, for example, refer to the sources below, which show that arthropods are a preferred dietary component of many birds living in a variety of habitats.
- Nyffeler, M.; Şekercioğlu, Ç. H.; Whelan, C. J. Insectivorous Birds Consume an Estimated 400–500 Million Tons of Prey Annually. The Science of Nature 2018, 105 (7–8), 47. https://doi.org/10.1007/s00114-018-1571-z .
- Manning, D. W. P.; Sullivan, S. M. P. Conservation Across Aquatic-Terrestrial Boundaries: Linking Continental-Scale Water Quality to Emergent Aquatic Insects and Declining Aerial Insectivorous Frontiers in Ecology and Evolution 2021, 9, 633160. https://doi.org/10.3389/fevo.2021.633160.
- Valkama, E.; Lyytinen, S.; Koricheva, J. The Impact of Reed Management on Wildlife: A Meta-Analytical Review of European Studies. Biological Conservation 2008, 141 (2), 364–374. https://doi.org/10.1016/j.biocon.2007.11.006.
- Razeng, E.; Watson, D. M. Nutritional Composition of the Preferred Prey of Insectivorous Birds: Popularity Reflects Quality. Journal of Avian Biology 2015, 46 (1), 89–96. https://doi.org/10.1111/jav.00475.
- Gerwing, T. G.; Kim, J.-H.; Hamilton, D. J.; Barbeau, M. A.; Addison, J. A. Diet Reconstruction Using Next-Generation Sequencing Increases the Known Ecosystem Usage by a Shorebird. The Auk 2016, 133 (2), 168–177. https://doi.org/10.1642/AUK-15-176.1.
L88: Also cite Seress et al. 2018
Seress G, Hammer T, Bókony V, Vincze E, Preiszner B, Pipoly I, Sinkovics C, Evans KL, Liker A. Impact of urbanization on abundance and phenology of caterpillars and consequences for breeding in an insectivorous bird. Ecological Applications. 2018 Jul;28(5):1143-56.
Answer:
Following the Reviewer's recommendation, the proposed citation has been added in the indicated place (in the new version in Line 91, references position: 50).
L93: Also cite Seress et al. 2020 and Grabarczyk et al. 2021
Seress G, Sándor K, Evans KL, Liker A. Food availability limits avian reproduction in the city: An experimental study on great tits Parus major. Journal of Animal Ecology. 2020 Jul;89(7):1570-80.
Grabarczyk EE, Gill SA, Vonhof MJ, Alabady MS, Wang Z, Schmidt JM. Diet composition and diversity does not explain fewer, smaller urban nestlings. Plos one. 2022 Mar 1;17(3):e0264381.
Answer:
We also agree with the reviewer's comment and recommendation. The proposed citation has been added in the indicated place (in the new version in Line 96, references position: 57-58)
L126: “praying blue tit”, do you mean Eurasian blue tit?
Answer:
That's what it was about Eurasian blue tit. The name has been corrected to the correct one (in the new version in Line 129).
Methods
L230-243: Describe here how landscape variables (first reported in L336-342) were calculated (for example, what program did you use for analysis, etc.).
Answer:
Thank you for pointing out this lack, some information about the use of the "QGIS Desktop 3.4.13" program and recording data in the GIS system was recommended, but it was a bit inaccurate and this information was missing the characteristics of the study area, and collecting information about its landscape structure. Currently, we have the following text “In addition, the distance of the centre of each meadow to the riverbed (m) and the edge of the nearest forest (m) and the length of the boundary line of each meadow occupied by shrubs and trees (m) were measured. These variables were measured in "QGIS Desktop 3.4.13" [146] using a loaded orthophoto from "Geoportal 2" (https://mapy.geoportal.gov.pl/), as well as measurements taken in the field using a Leica 1600 rangefinder and a rangefinder integrated with Leica Geovid R 15x56 binoculars”. (in the new version in Lines 235-240)
L281: Clarify what you mean by “GPS space”.
Answer:
Maybe it's too obvious and we used a mental shortcut. The abbreviation GPS stands for Global Positioning System. The locations of the nests were marked in the GPS system using the Garmin GPSmap76S and eTrex Vuista HGx. The collected navigation data were then downloaded to a computer and displayed against the orthophotomap using the "QGIS Desktop 3.4.13" program.
We added the information in the new version in lines 280-282.
L282-308: More information is needed regarding the nestling population and safe handling practices. Report the range of nestling ages included in the BCI. Did you typically find a nest with chicks or eggs? If eggs, how often did you go back to check on the nest and if/ once hatched, were nests checked more than once? What is the blood volume to nestling mass that was collected? How long were chicks handled? What steps were taken to ensure no premature fledging?
Answer:
Thank you for this comment. Indeed, some information was missing, but it seemed obvious or unnecessary. But as for the clarification of the ethical aspects, we can say that 1/ the research was conducted with the permission of the relevant Ethical Committee for animal experiments Animal and Regional Directors of Environmental Protection, 2/ the biological material (blood samples)was collected by one person, Jacek J. Nowakowski, who is also a veterinarian and an ornithologist with a long experience of working with birds, especially biometric research, 3/ all persons have received training on handling wildlife during experiments, and on the aspect of planning and carrying out such experiments with animals, 4/ all persons involved in the field work are ornithologists with extensive professional experience, certified to ring birds, 5/ a procedure has been developed, of course, which is actually standard today when it comes to collecting biological material from birds in the form of blood (work with up to 7 chicks in a brood is carried out very quickly, in addition, when removing chicks from the nest for ringing, there were always some chicks left in the nest, so that adult birds do not accidentally find an empty nest and so as not to leave traces.
In the case of the question about the frequency of nest inspections – conducting research by mapping method and observing also behavioral aspects of adult birds, observations of their foraging site, etc. in a species that quite variably, but usually not very strongly hides nests, they were quite quickly located by us during field work. Detected at various stages of brood development. The principle we applied was to harm the birds as little as possible, and knowing the basis of the element of breeding biology and behavior of the species, it was possible to recognize that, for example, the birds are building a nest, that we detect the nest at the stage of laying and predict when the chicks will appear and when it will be possible to ring them and collect research material. It can be said that there was usually only one check at the chick stage, and the birds were more or less at day 5-10 of development. The procedure itself for blood collection in passerines is based on the puncture of the cutaneous ulnar vein, and it is a small drop of blood from which the material is collected in one haematocrit microcapillary and glucose microcapillary and by touching with the edge of the slide a smear is spread from the droplet. This information was provided in the manuscript It is difficult for me to answer the question of what it looks like by weight, (ca. 50μl) but the procedure of this study is completely safe, we work with sparrows, tits, warblers.
We supplemented the description of the method with the following text below. In ty paragraph we have also included information on the number of nests found at the laying or nestling stage and the assessed breeding rates that the Reviewer also asked for. Previously, we only provided the number of nests and chicks used in this study. We did not collect blood from very small birds, as well as we gave up in the situation of detecting chicks close to leaving the nest, or it was not possible to control due to the organization of the study, so it did not seem relevant to include this information in the paper. But in accordance with the Reviewer's wishes, abbreviated are given in lines 284-294 (new version of manuscript).
“During the study period, we found 145 nests with brood. The average clutch size was 5.0±1.16 eggs/nest and 3.9±2.1 chicks/nest). In extensively used meadows, 32 nests were found (4.9±1.01 eggs/nest; 3.6±2.16 chicks/nest), in the vicinity of meadows fertilized with mineral fertilizers, 70 nests were found 4.9±1.26 eggs/nest, 3.9±2.13 chicks/nest), and in meadows fertilized with manure, 43 nests (5.1±1.05 eggs/nest; 4.2±2.01 chicks/nest). The chicks from which samples were taken were approximately 5-10 days old. Nests were found at various stages of breeding, which resulted from observations of birds' behaviour in their territories, but the procedure was focused on a single inspection of the nest at the fledgling stage to limit the possibility of nests being detected by predators. The date of the nest inspection was estimated so that it would be possible to ring the chicks, take measurements and take biological samples”.
.”
L310: Was body condition or hematological measures related to/correlated to nest of origin? For example, were nestlings in one nest in better condition that nestlings in a different nest but same habitat? If so, the parents may have different levels of experience rearing offspring/higher fitness, which could affect nestling condition. Please report either 1) that there is no correlation between nestling and brood or 2) run a GLM and include nest identity as a random effect.
Answer: This is very difficult to get an answer to. I considered mixed models with a random factor, but I concluded that the essence of the study is precisely the variation of the studied variables between nests. Birds within a nest should have similar parameters, since we expect variation caused by the effect of spatial distribution of nests and the influence of environmental factors. Of course, I cannot reject the hypothesis raised by the Reviewer that there may be some other effect superimposed on the study. It is difficult to maintain a fully controlled experimental setup under field conditions. Mixed methods are mainly used to remove some of the information, and excessive variability in the explained variable, and usually a random factor is introduced into the model to exclude some of the variance or control for possible autocorrelation, some of the covariance or to account for repeated observations. Here, I'm not sure that excluding nested-related variability is entirely appropriate. However, I made generalized linear mixed models as suggested and decided to include them in the manuscript because they do not change the nature of the inference. I introduced the id: nest and year of study as random variables. In the case of two models, it was inefficient to test with the id: nest factor I left the models with one random factor. Tables 2-6 have now been revised (lines 430-470) and minor corrections have been made to the results of the models in lines 411-412, 415-417.
L343: Report AIC values in a supplemental table. What was the AIC value used to assess model fit (assuming you used AIC and not weight to compare models).
Answer: The values of the AIC function were given next to each model, these were the final values for the selected model, which were model sorted in the algorithms of the Statistica 13.3 program. Since the algorithm for finding the best-fitting model is based on the method of highest reliability, different sets of variables are tested and the model with the smallest value of the Akaike criterion is selected. Now in Tables 2-6 (lines 43-470), I have given the values of the AICc criterion - a corrected Akaike criterion, implicit because it corrects for small numbers, and with large N equals AIC. Since I did the calculations GLMMs in SPSS 29.0 program, the default value of AICc is calculated, since the criterion does not need to be corrected in the situation of small samples or the program allows to estimate the Bayesian criterion. Here I did not comply with the request, as to whether it is reasonable to show many models and the AIC delta criterion. The estimation of deltaAIC or deltaAICc is mainly used in ranking models, mainly in testing the systems compared with the null model, which is built a priori by some assumptions of the variables, and in the comparative systems, the variables of the test (interesting to study) are introduced. AICc weighting is the best way to compare models and represents the relative probability of the model, so it is easy to interpret. The delta criterion requires a certain assumption and takes an a priori step, which is to carefully construct hypotheses to test. AIC should only compare models that interest us. In our case, we did not set up the hypothesis in such a way.
I have only included Fig. 1SA as an appendix to supplementary materials informing the purpose of PCA analysis.
Results
Report the sample size for the number of nests and chicks in each of the habitat types. Also report the mean and range of chick age at time of collection. For many nestlings, growth asymptotes as they reach a certain age. Did you collect samples before or after this time period? If both, please account for that in your models.
Answer:
As mentioned above, the information was included in lines 43-470 in the new version of the manuscript.
L352: Rather than “determined by” consider “driven by”.
Answer:
Was corrected.
Figure 2: In the figure description, include symbols used on the y-axis. For example… “Variation in abundance (n).” Also consider using Figure 2A, 2B, 2C labeling.
Answer:
As suggested by the Reviewer, an explanation of the symbols a, b, ab was added to the figure description, similar to Table 1.
L394: Clarify what you mean by, “pairs of birds nutritionally associated with…”.
Answer:
General information was provided in the Methods (Lines 283-284), but now it has been clarified in the indicated place. It will probably be more readable. Corrected according to the Reviewer's recommendation. Added explanation to the phrase “pairs of birds nutritionally associated with…”. The following content has been added in brackets “(feeding in meadows with extensively used)”.
Discussion
L443-448: Presence of beetles and orthoptera could also be slower to decay or decay less than soft bodied arthropods.
Answer:
Corrected according to the Reviewer's recommendation. A fragment emphasizing the selectivity of food found in pellets has been added, along with a citation.
However, it is worth remembering that examining the composition of pellets has its limitations related to the rate of digestion of various groups of insects, which is why the number of identified soft-bodied arthropods, e.g., is often lower. flies or spider fragments [171].
L469-480: Would be helpful for the reader if you related this information back to your topic sentence.
Answer:
The paragraph now in lines 511-528 was transformed.
L483: “arthropod activity” – please explain what type of activity you are referencing (foraging? Presence in an area?).
Answer:
Was corrected – (now Lines 530-531).
L497-503: This paragraph is a single, long sentence. Either combine this idea in a paragraph elsewhere in the discussion or further develop this idea into a full paragraph. Same comment for lines 520-525 and the three incomplete paragraphs in the conclusions.
Answer:
In line with the comments of the second Reviewer, the sentence was deleted. As recommended, the indicated fragments have been rephrased and integrated with other paragraphs.
Kind regards
Jacek J. Nowakowski
Round 2
Reviewer 1 Report
Comments and Suggestions for Authors
Accept in present form.
Author Response
Dear Reviewer, we would like to thank you once again for your evaluation of our manuscript and thank you that the explanations of inaccuracies, additions and proposed changes have found your acceptance.
Kind regards
Jacek J. Nowakowski
Reviewer 2 Report
Comments and Suggestions for Authors
Your revision looks fine to me.
In my original report I made three comments that were ignored in the revision. These are still relevant so I include them below with the new line numbers.
Line 237. Can you please describe – in a few words – the meaning of the terms ‘shrub’ and ‘scrub’? I am not clear on what type of vegetation is involved and how the two types differ.
Line 312. What does pitch mean here? Is it wing length?
Line 566. I think that this word should be ‘injured’ rather than ‘uninjured’. Otherwise, I am confused as the preceeding sentence describes insects being damaged by mowing. These are the ones that are available to the shrikes I assume.
Comments on the Quality of English LanguageThe English looks mostly fine.
Author Response
Dear Reviewer, we would like to thank you once again for your evaluation of our work and thank you that the explanations of inaccuracies, additions, and proposed changes have found your acceptance.
We are sorry to have omitted these three comments, even though there are several authors. They are indeed important and I sincerely thank you for your careful consideration of the text of the manuscript.
I set out the responses below
Line 237. Can you please describe – in a few words – the meaning of the terms ‘shrub’ and ‘scrub’? I am not clear on what type of vegetation is involved and how the two types differ.
We have deliberately used the two terms “scrub” and “shrub”, which denote slightly different types of vegetation. We have used the word “scrub” to describe the vegetation created by trees in the process of succession (young trees, e.g. Populus tremula, Alnus glutinosa, Betula sp., which start to encroach on grassland, wetlands). In the case of extensively used grasslands, this usually happens when livestock have not grazed them very often. At this point, reference should be made to the source papers, in which the authors indicate that the predominant wetlands, which were usually a mosaic of natural vegetation types, were transformed into grasslands – multi-cut meadows, in the process of which young tree clusters were removed. The word “shrub”, on the other hand, has been used to describe a typical shrub, as well as a vegetation type where the dominant woody elements are shrubs growing up to 5 meters in height at maturity (willow Salix sp., elder Sambucus nigra, etc.).
As we refer to the source papers we propose to leave both phrases. These terms are often used precisely to characterize such vegetation forms.
Line 312. What does pitch mean here? Is it wing length?
Thank you for pointing out the inaccuracy in this sentence. This is our error in this sentence - it is incorrectly phrased and does not contain de facto proper noun. It refers to tarsus length, which was measured with a calliper whose graduation (pitch) was to the nearest 0.1 mm. This has now been corrected.
Line 566. I think that this word should be ‘injured’ rather than ‘uninjured’. Otherwise, I am confused as the preceeding sentence describes insects being damaged by mowing. These are the ones that are available to the shrikes I assume.
Thank you for the reviewer's vigilance and careful review of the manuscript. Obviously, in the context of this aspect under discussion, the reference was to injured animals. The word injured should have been used.
This has been corrected.
Best regards
Jacek J. Nowakowski